# Ventromedial hypothalamus (VMHvl) nNOS neurons regulate social behaviors in a sex-specific manner

Vinícius Elias de Moura Oliveira [1,2] ✉, Ioana Bodea[1] & Julie Bakker [1] ✉

Neuronal nitric oxide synthase (nNOS) neurons are ubiquitously spread in the mouse brain. Data using knockouts and pharmacology have revealed that nNOS is essential for the display of sexual and aggressive behavior. Yet, the specific neuronal populations regulating those behaviors remain elusive. Here, we aim to study the role of the ventromedial hypothalamus (VMHvl)-nNOS neurons in social behaviors in both sexes. First, we evaluate whether the expression of nNOS overlaps with the well characterized estrogen receptor alpha (ERα + )-VMHvl population. Next, we assess how different social stimuli affected VMHvl-nNOS neurons' activity. Lastly, we use transgenic mice and viral approaches to ablate VMHvl-nNOS neurons and evaluate their impact on behavior. Our findings suggest that nNOS neurons constitute a small cluster within the VMHvl-ERα+ population that regulates social behaviors in a sex-specific manner. In males, those neurons seem to be essential for aggression, whereas in females for sexual behavior and social motivation.

Social behaviors such as mating, aggression, and prosocial approach are considered innate behaviors as they are displayed readily without the need for previous experience[1–3]. In general, those behaviors rely on stereotypical motor patterns and are embedded in developmentally hardwired neural circuits[2], which are known to exhibit strong sex differences[2–6] and are shaped by internal and external factors[1,4,6] such as reproductive states[2–5,7] and social experience[1,6,8–10], respectively.

Particularly hypothalamic areas have been strongly associated with the display of instinctive behaviors such as mating[3,11,12] and fighting[4,13,14]. Of note estrogen receptor α (ERα) and progesterone receptor (PR)-positive neurons (expressed in a ratio of 1:1[15,16]) in the ventrolateral part of the ventromedial nucleus of the hypothalamus (VMHvl) are crucial for the display of aggressive and sexual behavior in both sexes[15–19]. Although the role of ERα-PR neurons (accounting for 40–50% of total neurons in the VMHvl) on sociosexual behaviors has been extensively studied, the participation of other neuronal populations within the VMHvl, which is a highly heterogeneous and sexually differentiated region[20], remains understudied.

Of note, neuronal nitric oxide synthase (nNOS) neurons are found in the VMHvl[21] and are known to co-express PR in female mice[22]. Furthermore, compelling evidence has tied nNOS expression and possibly nitric oxide (NO) synthesis to a sex-specific regulation of social behaviors. Briefly, nNOS knockout (KO) males exhibit abnormal sexual behavior (excessive mounting) and exaggerated aggression compared to wild types (WT)[23,24] whereas females showed reduced lordosis[25] and maternal aggression[26].

Pharmacological evidence further demonstrated the role of nNOS activity and possibly NO signaling in social behaviors. In male mice, pharmacological inhibition of nNOS activity increased and decreased aggression depending on factors such as social experience[24,27] and synaptic plasticity[28]. In contrast to KO studies, inhibition of nNOS activity decreased sexual behavior in naive male rats without affecting sexually experienced rats[29]. On the other hand, in females, KO and pharmacological studies seem to be aligned as inhibition of VMHvl-nNOS neurons[22] or nNOS activity[30–32] in the VMHvl both reduced lordosis. Yet the role of nNOS neurons in virgin female aggression[4] as well as the participation of specific nNOS populations in various social behaviors such as social motivation, mate choice, mating, and fighting in a sex-specific context remains elusive.

Here, we first assessed whether nNOS expression in the VMHvl overlaps with the well-characterized ERα population in a sex-specific manner. Next, we exposed mice of both sexes to different social stimuli in order to trigger different behavioral responses and measured co-expression of nNOS and neuronal activity markers with different temporal dynamics, i.e., c-Fos (mating) and pERK (fighting), taking into account the effect of reproductive states and social experience. Finally, in the last set of experiments, we performed loss-of-function experiments using nNOS::cre mice and viral injections to specifically ablate nNOS neurons within the VMHvl and measure its consequences on behavior as well as used NO donors to rescue the behavioral deficits induced by deletion.

[1]Laboratory of Neuroendocrinology, GIGA-Neurosciences, University of Liege, Liege, Belgium. [2]Institute of Pathophysiology, University Medical Center of the Johannes Gutenberg University Mainz, Mainz, Germany. ✉e-mail: demourav@uni-mainz.de; jbakker@uliege.be

## Results

### VMH-vl nNOS neurons showed a sex-specific co-expression with Erα

We confirmed the sex-dimorphic expression of ERα in the VMHvl, i.e., females show a higher number of ERα[+]-cells (Fig. 1B)[16,18,33]. Additionally, we confirmed previous reports on the absence of sex differences in nNOS neurons within the VMHvl[21]. Nevertheless, we found some sex differences when looking into the co-expression of those two proteins. In females, around 60% of all nNOS neurons co-expressed ERα compared to 40% in males (Fig. 1A, Supplementary table 1). Taking together these data revealed that nNOS neurons do not overlap completely with the well-characterized PR/ERα-VMHvl population[1,2,5,11]. They rather constitute a subpopulation of ERα neurons similar to other genes, such as cckar[33,34] or npy2r[33,35] populations. Additionally, these data showed that approximately half of nNOS neurons co-express estrogen receptors, which have been shown to be essential for the display of social behaviors in rodents[1,5,36].

Finally, we noticed that most nNOS and nNOS-ERα neurons were localized in the posterior-lateral part (pvll) of the VMHvl of females (Fig. 1C), a subregion present only in female mice that has been linked with female sexual but not aggressive behavior[16,34,35].

### VMH-vl nNOS neurons exhibit a sex-specific activation pattern after same- and opposite-sex social stimulation

Next we investigated whether different social stimuli recruit VMHvl-nNOS neurons in a sex-specific manner (Experiment A and Fig. 2A).

Regarding females, measuring aggressive behavior in virgin female mice is challenging[4], as virgin female C57 mice do not readily display aggressive behavior[4,16]. Thus, we decided to use two animal models of exacerbated aggression as positive controls of heightened female aggression: lactating females[4,37] and IST females[1,4,38]. The IST protocol entailed high

although variable levels of aggression in a separate cohort of animals (Supplementary Fig. 1A and Supplementary Video 1); IST females also showed increased activation of oxytocin neurons in the paraventricular (PVN) and supraoptic nucleus of the hypothalamus (Supplementary Fig. 1B), OXT-c-FOS percentage of colocalization in the PVN also positively correlated with percentage of time spent on aggression (Pearson's Correlation PVN: $r = 0.867$, $p = 0.0005$; SON: $r = 0.528$, $p = 0.09$). Those results are in line with previous reports on oxytocin release having pro-aggressive effects in female rodents, independent of the reproductive state[4,37,39–41].

Regarding males exposed to mating and fighting, we found that around 11% of the nNOS neurons responded to both social stimuli indistinctly, whereas 15% of the neurons colocalized only with pERK (putatively fighting neurons) and 19% colocalized only with c-FOS (putatively mating neurons). Those results showed that around half of VMHvl-nNOS neurons respond to both social stimuli and that those neurons seem to segregate into different neuronal populations (Fig. 2B–D, Supplementary Table 1).

Females exposed to males (who displayed lordosis behavior) showed an increased colocalization of nNOS and cFOS in the VMHvl compared to animals exposed to an object (on average, 14% of nNOS neurons were recruited). Interestingly, exposure to a same-sex juvenile did not increase neuronal activity in VMHvl-nNOS neurons of females (Fig. 2E, F). Accordingly, neither IST females nor lactating females showed any differences in the colocalization of nNOS and pERK in response to an aggressive encounter (Fig. 2G). Taken together, these data suggest a sex difference in how nNOS neurons are recruited by social stimuli.

Importantly, most results using the percentage of activated nNOS cells remained when we compared the total number of colocalized cells between groups. Interestingly, the total number of nNOS neurons in the VMHvl tended to differ between behavior-exposed and object-exposed males (Supplementary Table 1), with behaviorally exposed mice tending to exhibit

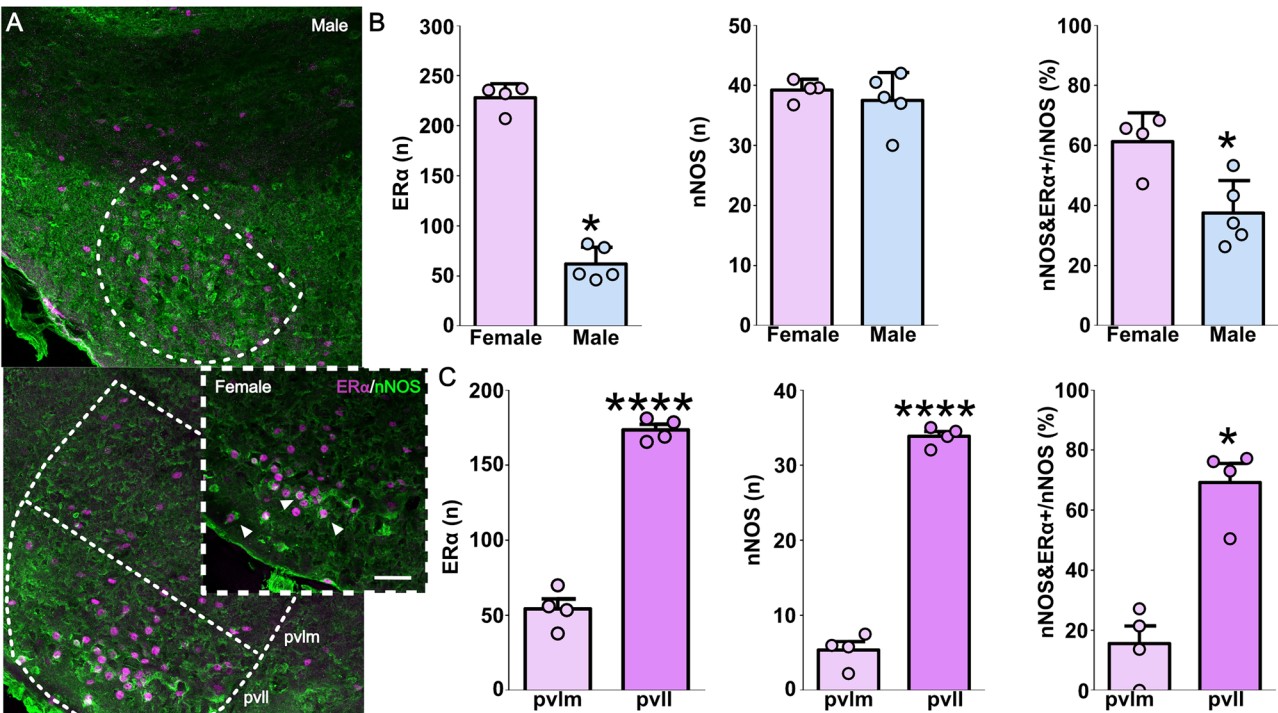

**Fig. 1 | VMH-vl nNOS neurons exhibit a sex-specific co-expression with ERα.**
Maximal z-projection of ERα (magenta, Alexa-fluor-546 anti-rabbit) and nNOS (green, Alexa-fluor-488 anti-goat) in the ventrolateral portion of the ventromedial hypothalamus (VMHvl) in male and female mice, inset (dotted lines) in the lower panel of the female VMHvl shows colocalization of both markers (**A**). ERα was more abundant in the VMHvl of females compared to males (Mann-Whitney U test U = 0.0, $p = 0.015$), whereas nNOS neurons did not differ between sexes (two-tailed Student's t-test $t_{(7)} = 0.69$, $p = 0.512$). Consequently, co-expression of nNOS and ERα was sexually dimorphic with a higher proportion of nNOS neurons expressing ERα in females ($t_{(7)} = 3.44$, $p = 0.01$) (**B**). Most nNOS neurons were located in the lateral part of the VMHvl (pvll) instead of the medial portion (pvlm) (nNOS: $t_{(6)} = 22.02$, $p < 0.0001$; nNOS-ERα/nNOS: U = 0.0; $p = 0.028$) (**C**). Data are presented as mean + s.e.m. *$p < 0.05$; ***$p < 0.001$; ****$p < 0.0001$. Scale bar 100 μm. Females $n = 4$ and males $n = 5$.

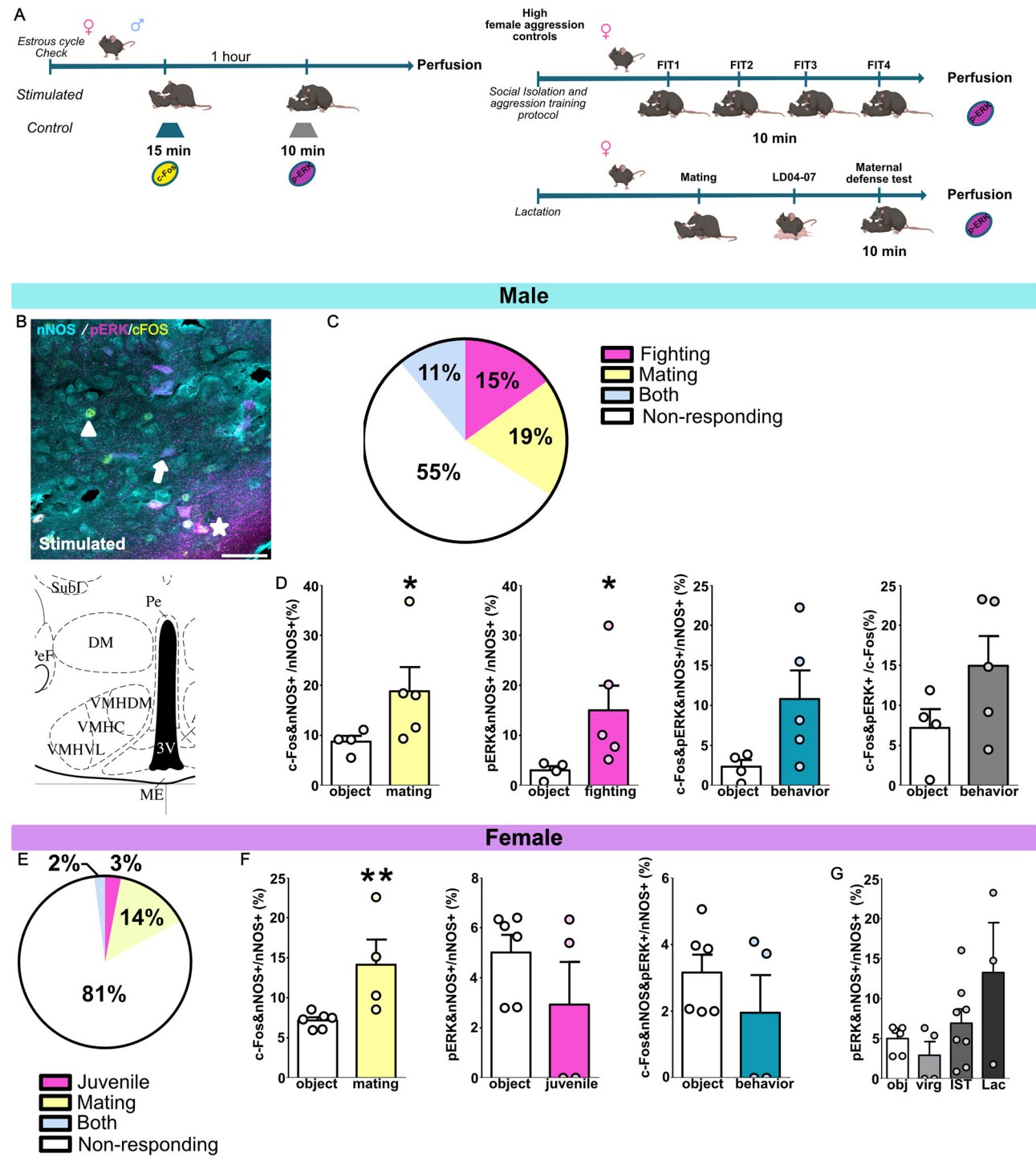

fewer nNOS neurons. Additionally, the number of c-FOS and nNOS cells did not differ for males when cell numbers were used for comparison. Although unexpected, this fits the observation that ablation of nNOS neurons only mildly affects male sexual behavior (Fig. 3).

## Ablation of VMH-vl nNOS neurons strongly reduces aggression and mildly affects sexual behavior in males

To confirm the involvement of nNOS neurons in different aspects of social behaviors, we ablated VMHvl-nNOS neurons and measured different social behaviors in males (Experiment B and Fig. 3A).

We confirmed that nNOS deleted (nNOS-del) mice had a 3-fold decrease in the number of nNOS neurons in the VMHvl compared to

controls (Fig. 3F). Regarding behavioral data, ablated mice exhibited a lower percentage of time on aggression, threat, and attacking the intruder. Additionally, nNOS ablation almost suppressed attack bites as only 40% of the nNOS-del animals displayed attacks in at least one of the 3 RIs, whereas, in the control group, all animals attacked the intruder at least once. Consequently, nNOS-del mice showed a reduced number of attacks and an increased latency to attack the intruder (Fig. 3B). Finally, the percentage of time spent on aggression ($3^{rd}$ RI) correlated with the number of nNOS neurons in the VMHvl (Fig. 3B). Interestingly, the effect of nNOS deletion could not be compensated or reversed by re-engaging in aggressive interactions as nNOS-del animals did not show an increase in aggressive behavior after multiple RIs. Furthermore, reduced aggression could not be

**Fig. 2 | Same- and opposite-sex stimuli differentially recruit VMHvl-nNOS neurons in mice.** Schematic drawings depicting the experimental designs followed in Experiment A. Mice of both sexes were confronted either with an object (control) or an opposite-sex-conspecific for 15 min (mating). One hour later, those animals were again exposed to an object (controls) or a same-sex-conspecific (fighting) for 10 min after the test animals were perfused. High-aggression female controls consisted of isolated and aggression-trained (IST) females who underwent 4 consecutive female intruder tests and lactating females (lactating day, LD, 4-7) who were also confronted with an intruder for 10 min prior to perfusion (**A**) (Partially Created in BioRender. Bakker, J. (2025) https://BioRender.com/1kdxngi). Maximal z-projection of pERK (magenta, Alexa-fluor-546 anti-rabbit), c-FOS (yellow, Alexa-fluor-488 anti-guinea-pig), and nNOS (cyan, Alexa-fluor-633 anti-goat) in the ventrolateral portion of the ventromedial hypothalamus (VMHvl) in male and female mice, which were confronted with either an object (control) or a conspecific (stimulated) (**B**). In males, around 15% or 19% of nNOS neurons were exclusively activated by fighting by showing pERK colocalization (magenta pERK&nNOS[+]/

nNOS) or mating by showing colocalization with c-FOS (yellow,cFos&nNOS/nNOS[+]), respectively. Whereas 11% of the neurons showed a colocalization of cFos&nNOS&p-ERK/nNOS (white stars (**B**), blue bars (**D**). Thus, both mating or fighting increased the proportion of nNOS neurons active, showing either c-FOS (yellow bars in (**D**) and arrowheads in (**B**), Mann-Whitney U test U = 0.0, p = 0.031) or p-ERK (magenta bars in (**D**), and arrows in (**B**), U = 0.0, p = 0.015), respectively (**D**). In females, around 3% or 14% of nNOS neurons were exclusively activated by same-sex-conspecific/dominance (magenta) or mating (yellow), respectively. Whereas 2% of the neurons showed a colocalization of the three markers (**E**). Thus, only mating (yellow bars, c-FOS) increased the proportion of nNOS neurons active (Mann-Whitney U test U = 0.0, p = 0.009) (**F**). Finally, neither aggressive IST nor lactating females showed increased pERK colocalization with nNOS after being exposed to a same-sex intruder (Kruskal-Wallis test, K = 6.66, p = 0.157, (**G**) Data are presented as mean + s.e.m.*p < 0.05;**p < 0.01: Scale bar 100 μm. Males: obj n = 4 and Behavior n = 5; Females: obj n = 6 virg n = 4 Lac n = 3 and IST n = 8.

associated with a lack of social motivation as nNOS-del animals preferred social stimuli over an object (displaying social preference) (Fig. 3C) and did not show any effects on social investigation in the RI (Supplementary Table 2).

Ablation of nNOS neurons mildly affected mounts and intromissions as well as intromission latency when animals were tested for 10 min (Fig. 3D). Additionally, successive testing seemed to recover the effect of deletion as nNOS-del animals showed an increased number of mounts/intromissions on Test 3 compared to Test 1, consequently, there was no difference between nos-del and controls on Test 3, regarding those parameters (Fig. 3D). As males did not show a strong phenotype after being tested during short sexual behavior tests, we decided to challenge them by performing a 30 min session. Under those conditions, NOS-del mice displayed a lower number of mounts (p = 0.05) and intromissions as well as a longer latency to mount the receptive female (Fig. 3E). Accordingly, nNOS-del animals showed impaired mate preference, meaning that they did not prefer an estrous female over a male mouse (Fig. 3C). Taking together these data suggests that nNOS neurons might not be directly involved with the display of sexual behavior (consummatory behavior) but rather with female recognition (appetitive behavior) in males (Supplementary Table 2).

### Deletion of VMH-vl nNOS neurons impairs female sexual behavior and social motivation

Next, we performed similar experiments in female nNOS::Cre mice and evaluated different aspects of social interactions (Experiment C and Fig. 4A).

Neuronal ablation was confirmed as nNOS-del females showed a reduced number of nNOS neurons in the VMHvl compared to controls (Fig. 4F). Additionally, we did not find an effect of hormonal treatment or estrous cycle on the number of nNOS neurons (Supplementary Table 3). Surprisingly, we found strong sex differences in the role of nNOS neurons on social behaviors, matching our previous findings with neuronal activity. In contrast with males, neuronal ablation did not affect female aggressive/dominant behaviors toward a juvenile (Fig. 4B). However, social preference was abolished in nNOS-del females (Fig. 4C).

Concerning reproductive behaviors, nNOS ablation in the VMHvl nearly abolished lordosis behavior in females. Of note, successive behavior testing could not rescue the effect of neuronal ablation on lordosis. Accordingly, the number of nNOS neurons in the VMHvl correlated with the lordosis quotient (Fig. 4E). Finally, the effects of nNOS deletion in females could not be tied to an impairment in mate preference[42], as mate preference remained unchanged in nNOS-del females. In fact, those females preferred the male over the female even in the non-receptive phases of the estrous cycle (metestrus-diestrus) (Fig. 4C), indicating that ablation enhanced male preference in the absence of hormonal priming.

### Increasing NO availability did not rescue the effects of VMHvl-nNOS ablation

In the last set of experiments, we attempted to disentangle the role of NO and nNOS neurons in social behavior; to do so, we combined viral ablation, pharmacology, and behavior (Experiment D).

Deleted males displayed a reduced number of mounts & intromissions and attacks, as well as spent a lower percentage of time on aggressive behavior, those effects could not be rescued by the administration of SNAP-BAY. Females also exhibited reduced lordosis quotients, this phenotype could not be rescued by SNAP-BAY (Supplementary Table 4). Those results indicated that nNOS neuron deletion rather affects the social behavior circuit instead of NO neurotransmission in the VMHvl.

### Discussion

A myriad of behaviors ranging from social investigation and affiliative approach to mating and fighting emerge from interactions between conspecifics. Generally, those behavioral responses rely on several factors such as conspecific cues, internal states marked for physiological/hormonal alterations, and previous experiences[2,43]. In this manuscript, we investigated how VMHvl-nNOS neurons are affected by same- and opposite-sex conspecific cues as well as how ablation of those neurons affects the display of social behaviors itself.

Our data revealed that nNOS neurons in the VMHvl are recruited in a sex-specific manner by male and female cues. In females, those neurons were exclusively activated by mating (male cues) whereas in males both mating with a receptive female and fighting with a smaller male intruder increased the activation of VMHvl-nNOS neurons. Interestingly, those neurons were not activated in lactating females[4,37] or highly aggressive IST females[1,4,39,40], known to display exacerbated levels of aggression, indicating that the absence of activity of those neurons in females is not reproductive state, social experience, or aggression level-dependent. Furthermore, this is consistent with the majority of the nNOS neurons being located in the VMHvlpvl, tied to sexual but not aggressive behavior in females[16,38,44].

Surprisingly, the ablation of VMHvl-nNOS neurons in males nearly abolished aggressive behavior without affecting social preference. Although the loss of function data matches the neuronal activity after aggression, these data are rather unexpected. Previous studies using nNOS-KOs have shown that those animals exhibit exaggerated aggressive behavior[23,24]. Pharmacological inhibition of nNOS activity systemically also strongly increased aggression in male mice, particularly when those animals were single-housed[24,27]. Furthermore, nNOS enzymatic inhibition also decreased social investigation in single-housed mice[24]. Conversely, a recent study has found that inhibition of NMDAR-dependent PSD95/nNOS activity either i.p. or intracerebroventricularly (i.c.v.) strongly decreases social isolation-induced aggression in male mice, similar to our findings[28]. Pretreatment with L-arginine a precursor of NO prevented the effect of PSD95/nNOS inhibition on aggression, indicating those effects were NO-dependent[28].

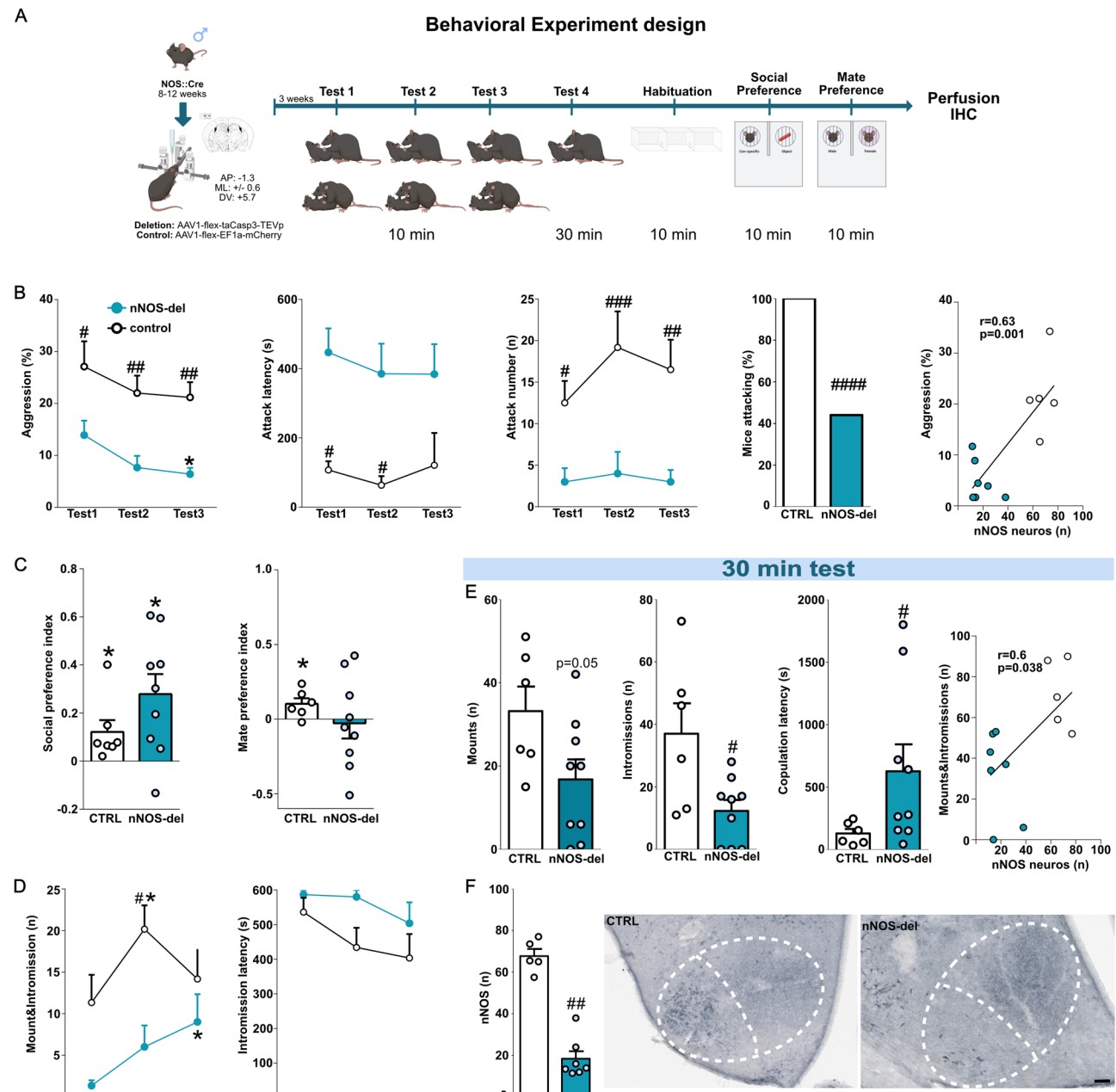

**Fig. 3 | Ablation of nNOS neurons in the VMHvl strongly reduced aggression and mildly affected sexual behavior in males.** Schematic drawings depicting the experimental designs followed in Experiment B. nNOS:cre (8–12 weeks) were injected bilaterally into the VMHvl with either AAV1-flex-taCasp3-TEVp (nNOS-del, blue) or AAV1-flex-EF1a-EGFP-WPRE (control, white) virus. Three weeks later, both groups were tested for sexual behavior for 10 min and directly after for aggressive behavior (10 min) on 3 consecutive days. After 4 days animals were again tested for sexual behavior for 30 min and 4 days later were tested in the three-chamber apparatus for social and mate preferences, before perfusion (**A**) (Partially Created in BioRender. Bakker, J. (2025) https://BioRender.com/xvhcfhy). Deleted animals (nNOS-del, blue) showed a decreased percentage of time spent on aggression (Virus effect: $F_{(1, 13)} = 28.55$, $p = 0.001$; Training effect: $F_{(2, 26)} = 3.62$, $p = 0.04$) which was reflected in a higher latency to attack (Virus effect: $F_{(1,13)} = 19.58$, $p = 0.0007$), lower number of attacks (Virus effect: $F_{(1,13)} = 37.11$, $p < 0.0001$) as well as a lower proportion of animals attacking (Fisher's exact test, $p = 0.0001$). Consequently, aggression in the third test positively correlated with the number of nNOS neurons in the VMHvl (Pearson's correlation $r = 0.636$, $p = 0.001$) (**B**). Social

preference was not affected by neuronal ablation (One sample t-test ctrl: t-test $t_{(6)} = 2.49$, $p = 0.047$; nNOS-del: $t_{(5)} = 3.35$, $p = 0.01$), however, mate preference was absent in nNOS-del males (ctrl: $t_{(5)} = 2.49$, $p = 0.04$; nNOS: t-test $t_{(8)} = 0.26$, $p = 0.79$) (**C**). Regarding sexual behavior, the number of mounts and intromissions was decreased on test day 2 compared controls. However, nNOS-del mice showed an increase in mounts and intromission on test day 3 (Virus effect: $F_{(1,13)} = 9.242$, $p = 0.009$; Training effect: $F_{(2,26)} = 5.690$, $p = 0.008$) (**D**). Once animals were tested for 30 min, a reduction in sexual behavior was more evident in nNOS-del mice, which displayed fewer mounts (two-tailed Student's t-test $t_{(13)} = 2.147$, $p = 0.05$) and intromissions (two-tailed Student's t-test $t_{(13)} = 2.759$, $p = 0.01$) (**E**). Additionally, the latency to mount a female was higher (U = 10, $p = 0.049$). Mounts and intromissions also positively correlated with the number of nNOS neurons in those animals ($r = 0.3634$, $p = 0.038$) (**E**). Finally, deletion was confirmed as nNOS-del mice showed a lower number of nNOS neurons (U = 0.0, $p = 0.002$) in the VMHvl compared to controls (**F**). Scale bar 100 μm. *$p < 0.05$ vs. test or 0.0 (preferences); #$p < 0.05$,##$p < 0.01$;###$p < 0.001$ vs. control. Data are presented as mean + s.e.m. Behavior: Ctrl $n = 7$nNOS-del $n = 9$; Histology: Ctrl $n = 5$, nNOS-del $n = 7$.

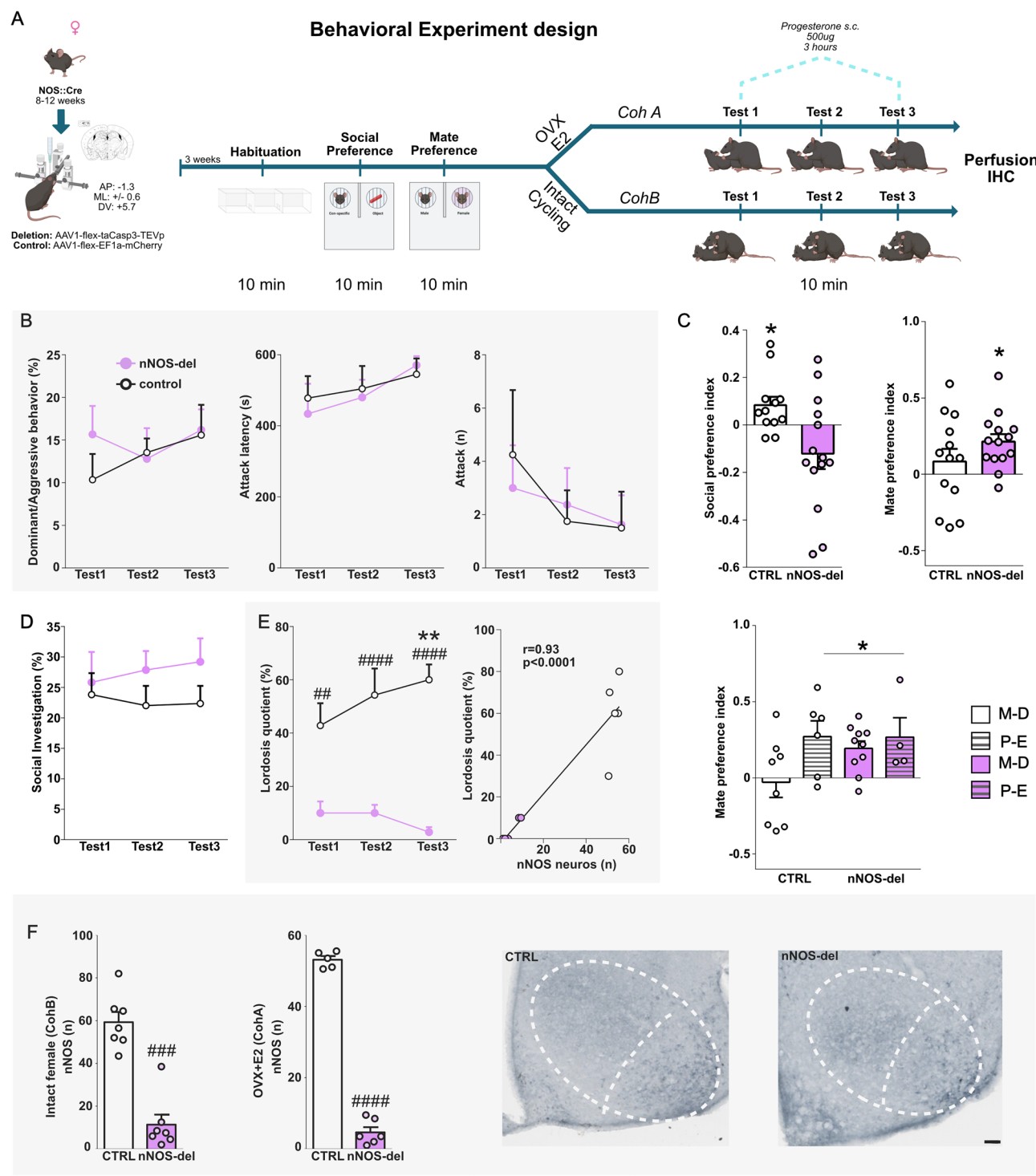

Here one should differentiate between the role of nNOS-positive neurons and nNOS enzymatic activity and consequently NO neurotransmission on aggression. Early, pharmacological[24,27] and KO[23] studies particularly focused on the latter whereas our study is centered on the neuronal populations expressing nNOS. In fact, elevating NO levels via the administration of a NO-donor failed to rescue the behavioral deficits seen in aggressive behavior in our study. This suggests that the effect on nNOS neuronal ablation is not mainly underlied by NO-transmission but rather by their role in the neuronal circuitry of aggression. Alternatively, another possible explanation would be that VMHvl-nNOS neurons are particularly engaged in the plasticity-dependent nNOS activity similar to the study of

Yang et al.[28]; in fact, plasticity-dependent enhancement of aggressive behavior has been tied to the VMHvl in male winners[9]. Future studies should verify that hypothesis.

Regarding male sexual behavior, our data revealed a mild effect of nNOS neuron deletion in reducing the number of intromissions and the copulation latency when animals were tested for 30 min. On the other hand, multiple tests seemed to rescue the ablation effect on male sexual behavior partially. In line with our observations, a study performed in male rats using L-NAME (an inhibitor of nNOS) i.c.v. has shown that nNOS inhibition reduces mounting and abolishes ejaculation in naïve but not experienced rats[29], indicating sexual experience might overrule the effects of NO on

**Fig. 4 | Ablation of nNOS neurons in the VMHvl strongly reduced sexual behavior and abolished social preference in female mice.** Schematic drawings depicting the experimental designs followed in Experiment C. nNOS:cre (8–12 weeks) were injected into the VMHvl bilaterally with either AAV1-flex-taCasp3-TEVp (nNOS-del, purple) or AAV1-flex-EF1a-EGFP-WPRE (control, white) virus. Three weeks later, both groups were tested in the three-chamber apparatus for social and mate preferences. Next, animals were split into two Cohorts, cohort A underwent ovariectomy (OVX) and was implanted with Silastic capsules containing crystalline 17-β-estradiol (diluted 1:1 with cholesterol) subcutaneously. Those animals were treated s.c. with 500 µg progesterone 3 h prior being tested for sexual behavior. Cohort B was kept ovary intact and was tested in the female intruder test, by being confronted with a juvenile (p16-21) CD1 mouse (**A**) (Created in BioRender. Bakker, J. (2025) https://BioRender.com/fgrvx01). Deleted (nNOS-del, purple) and control (white) females showed very mild levels of aggression/dominance, which were not affected by neuronal ablation (**B**). Mate preference, on the other hand, remained unchanged in nNOS-del females independently of the estrous cycle status (ctrl P-E: t-test $t_{(5)} = 2.63$, $p = 0.046$; ctrl-MD: t-test $t_{(7)} = 0.28$, $p = 0.28$; nNOS-del P-E: t-test $t_{(3)} = 0.2093$, $p = 0.12$; nNOS-del-MD: t-test $t_{(9)} = 3.971$, $p = 0.003$ P-E: proestrus/estrus phase of the estrous cycle; M-D: metaestrus/diestrus phase of the estrous cycle) (**C**). However, social preference was abolished in nos-del animals (One sample t-test ctrl: t-test $t_{(11)} = 2.345$, $p = 0.038$; nNOS-del: $t_{(13)} = 1.87$, $p = 0.08$), this could not be linked to social approach as social investigation deficits in the FIT did not differ between groups (**D**). Female sexual behavior was strongly disrupted by the deletion of VMHvl-nNOS neurons, as females displayed a reduced lordosis quotient (Virus effect: $F_{(1,12)} = 34.29$, $p < 0.0001$; Training effect: $F_{(2,24)} = 1.305$, $p = 0.29$; Virus x training effect: $F_{(2,24)} = 4.969$, $p = 0.01$), which negatively correlated with the number of nNOS neurons in the VMHvl (Pearsons correlation r = 0.872, $p < 0.0001$) (**E**). Finally, deletion was confirmed in both cohorts as nNOS-del females (intact and OVX-E2) showed a lower number of nNOS neurons (CohA (OVX + E2) U = 0.0, $p = 0.0043$; CohB (intact) U = 0.0, $p = 0.0006$) in the VMHvl compared to controls (**F**). Scale bar 100 µm. *$p < 0.05$, **$p < 0.01$ vs test or 0.0 (preferences); #$p < 0.05$,##$p < 0.01$; ###$p < 0.001$; ####$p < 0.0001$ vs control. Data are presented as mean + s.e.m. Behavior: Ctrl n = 8 nNOS-del n = 8; Histology: Intact: Ctrl n = 7, nNOS-del n = 7; ovx+E2 Ctrl n = 5, nNOS-del n = 7.

mating. This mild effect of NO synthesis and VMHvl-nNOS neurons on male sexual behavior might arise from the participation of nNOS in the detection of female cues (appetitive phase) rather than copulation itself. Consistently, we have found that VMHvl-nNOS neurons are not strongly activated by female cues (Fig. 2). Additionally, nNOS-del males exhibited impaired mate preference and took longer to intromit and mount (Fig. 3). Thus, one could hypothesize that after deletion, males display less consummatory behaviors because they struggle to identify the receptivity cues of a potential mate. Accordingly, nNOS-KO studies have shown that males usually display persistent mounting even towards non-receptive (anestrous) females, compared to WT[23]. Taking together with our findings, this indicates that NO signaling and nNOS neurons might be involved rather in mate detection/choice than copulatory behaviors per se. Furthermore, other nNOS neuronal populations might be involved in male copulation, for example, sexually experienced rats showed increased nNOS mRNA in the PVN compared to naïve rats[29].

In contrast to males, VMHvl-nNOS neurons were only activated in females after mating with an adult male. Interestingly, exposition to a juvenile intruder which is known to trigger dominant behavior (mounts) in C57 female mice[38] as well as aggression displayed by lactating females[4,37] and IST females[4,39,40] did not increase pERK expression in VMHvl-nNOS neurons (Fig. 2), indicating that neither same-sex cues, dominant behavior, nor fighting recruits nNOS neurons in females. Accordingly, nNOS-del females did not show any differences in their levels of aggression&dominance (Fig. 4).

Accordingly, nNOS-KO virgin females do not show differences in the display of aggressive behavior[23]. Lactating nNOS-KO, on the other hand, showed a moderate reduction in maternal aggression[26]. Pharmacological studies in female prairie voles have shown similar, although milder effects. Inhibition of nNOS activity i.p. decreased maternal aggression after 2 but not 3 days of treatment[45]. Those results might indicate different mechanisms regulating the involvement of nNOS in aggressive behavior depending on the reproductive state. Another possible explanation is that other nNOS populations might be recruited during maternal aggression, indeed the number of nNOS neurons expressing citrulline (indirect NO synthesis marker) increased in lactating animals after an aggressive encounter, in regions known to regulated maternal aggression[46,47], such as the medial preoptic area (MPOA) and the PVN (where citrulline cells correlated with the display of aggressive behavior)[26,45].

We also found that nNOS-del females did not show a social preference, which is in agreement with previous data in males demonstrating that inhibition of nNOS activity decreases exploration of male urine[24]. Perhaps, the lack of dominant or aggressive behavior in virgin females arises from the fact those animals show low levels of motivation to interact and explore an intruder, although non-aggressive social investigation did not differ between control and nos-del females (Fig. 4D).

Finally, nNOS-del females showed a striking reduction in lordosis behavior (nearly abolished) which could not be linked to an impairment in mate choice (Fig. 4). Furthermore, in contrast to males, sexual experience did not rescue/compensate for the effect of VMHvl-nNOS deletion in females (Fig. 4). These data align with i) our neuronal activity data showing that VMHvl-nNOS neurons are active after mating in females (Fig. 2), the localization of nNOS neurons in the VMHvlpvl (Fig. 1) and previous reports of our group using nNOS-KO[25] and behavioral pharmacology[30], as well as from another group showing that inhibition of VMHvl-nNOS neurons via chemogenetics strongly reduces lordosis in mice[22]. Surprisingly, although the nNOS-KO phenotype could be reversed by elevating NO levels using SNAP-BAY[25], the same treatment failed to rescue the lordosis deficit of nNOS-del females. This again indicates that this ablation might affect the circuitry of female sexual behavior[11] rather than NO-neurotransmission. However, further investigation is needed in order to disentangle local NO production and nNOS activity from nNOS neuron activity, as i.p. injections could act in brain regions other than the VMHvl.

Surprisingly, nNOS-del diestrus females preferred a male over a female. This was unexpected, previous data from our lab shows that nNOS enzymatic activity is necessary for mate preference in females[25] additionally, male odors were able to activate nNOS neurons[31]. Knowing that i) PR neurons and nNOS neurons are co-expressed[22], ii) PR is co-expressed in nearly a 100% ratio with ERα in the VMHvl[15,16] iii) PR-VMHvl neurons are involved in mate rejection[19] and iv) PR signaling in the olfactory system silences male cue detection during diestrus, leading to an absence of male preference[48], one could hypothesize that by ablating nNOS neurons we partially eliminate some of the PR (ERα) neurons that might either be involved in mate rejection and/or recognition, therefore females start to show a male preference irrespective of the estrous cycle phase. Future studies should investigate the link between mate preference, VMHvl-nNOS, PR- and ERα signaling.

Finally, our data confirm previous findings portraiting the VMHvl as a hotspot for social behavior. Additionally, we characterized a novel neuronal population (nNOS neurons) from the behavioral point of view. Our data show that nNOS neurons in the VMHvl show sex differences in co-expression with ERα as well as how they respond to social stimuli. In females, these neurons seem to preferably respond to male cues whereas in males to both sexes. Finally, male VMHvl-nNOS neurons seem to be crucial for aggressive behavior and mate choice whereas in females those neurons seem to be essential for sexual behavior and social motivation (Fig. 5).

## Materials and methods
### Animals
Experiments were carried out in adult (8-12 weeks) male and female mice, which were either purchased from Charles River Laboratories (Les Oncins, France, CD1 and C57BL6/J) or bred in the animal facilities of the University

**Fig. 5 | VMHvl nNOS neurons modulate social behaviors in a sex-specific manner.** In males (right side, blue) nNOS neurons are activated by male and female cues and are scattered throughout the VMHvl additionally, those neurons are needed for the display of male sexual and aggressive behavior. In females (left side, purple), nNOS neurons are recruited by male but not female cues. Additionally, those neurons are mostly located in the lateral portion of the VMHvl, an area involved in female sexual behavior. VMHvl-nNOS neurons were found to be crucial for female sexual behavior and social preference. Created in BioRender. Bakker, J. (2025) https://BioRender.com/03tq6rp.

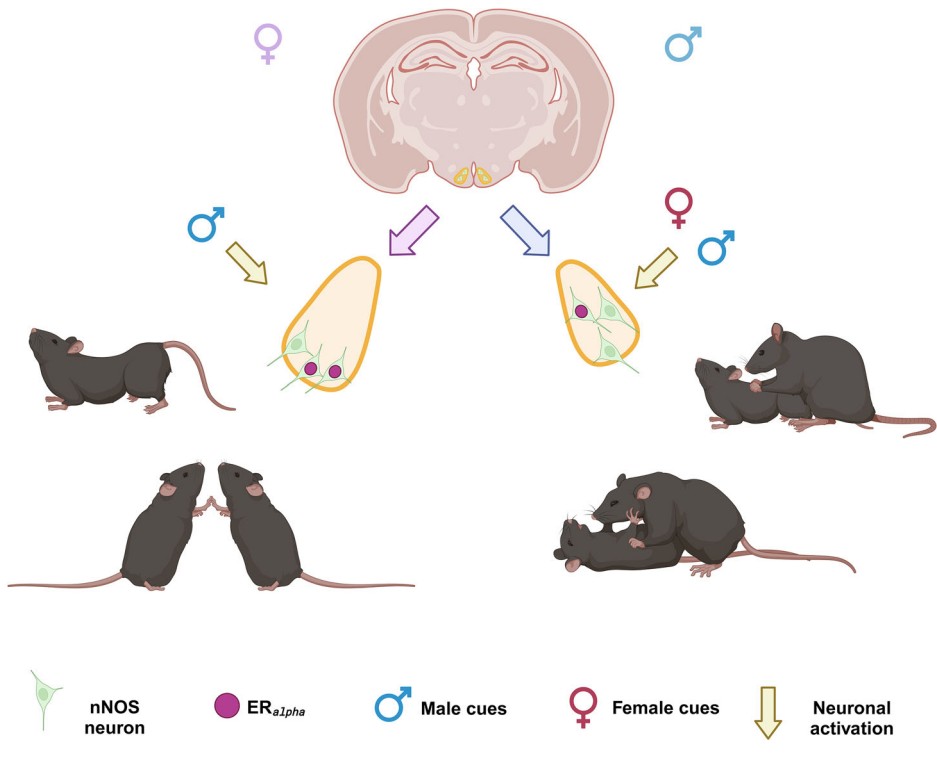

of Liège (nNOS::cre, Nos1cre/+, Nos1-ires-cre; B6.129-Nos1tm1(cre) Mgmj/J and WT, kept in C57BL/6 J background[49]). Viral ablation experiments were conducted in nNOS::Cre mice (Figs. 3, 4, and Supplementary Table 3) whereas immunohistochemistry experiments were conducted in C57BL6/J mice (Figs. 1, 2, and Supplementary Fig. 1). For aggressive behavior testing, adult male (5–6 weeks) and juvenile female (p16-21) CD-1 mice were used as intruders for male and female residents, respectively. For sexual behavior receptive (estrous) females (6–8 weeks) and experienced breeders (8–12 weeks old) C57BL6/J were used as stimuli to test male and female subjects, respectively.

All stimulus animals were obtained via Charles Rivers Laboratories (Les Oncins, France). Mice were kept under controlled laboratory conditions in ventilated shelves with (12:12 h light/dark cycle; lights off at 10:30 h, 21 ± 1 °C, 60 ± 5% humidity, standard mouse nutrition (RM3), and water ad libitum). Stimulus and experimental animals were housed in large mouse cages (42 x 32 x 18 cm) with sawdust bedding in groups of four. Importantly, after viral injection and prior resident or female intruder (RIT or FIT) tests, subjects were singly housed in individual mouse cages (33 x 16 x 13 cm) for at least one week to induce territoriality. We have complied with all relevant ethical regulations for animal use, all procedures described here were approved by the local Animal Ethics Committee of the University of Liège (protocol numbers 22-2469 and 2195).

### Aggressive behavior

All tests took place in the early dark phase under dim red light conditions. For the *resident intruder test* (RIT), male mice were confronted with an unfamiliar same-sex intruder for 10 min. Intruders were 10–20% lighter than residents[1,33,50]. Behavior was scored live and later via videos by an observer blind to the experimental condition using JWatcher event recorder Program[34,39]. As described elsewhere[39] the percentage of time spent on four major sets of behaviors was scored: (i) aggressive behavior, consisting of attacks, threats, chases, tail rattles, offensive grooming, offensive up-right; (ii) neutral behaviors, consisting of exploring (investigating the home-cage), drinking and eating, autogrooming, immobility; (iii) social behaviors (non-aggressive social interactions, sniffing, following); and (iv) defensive behavior (submissive posture, kicking a pursuing intruder with hind limb). In

addition, we scored the frequency of attacks as well as the latency of the first attack.

In the *female intruder test (FIT)*, females were typically confronted with a juvenile (p16-21) CD-1 female for 10 min, following a combination of the protocols described by Oliveira et al. 2021[39] and Hashikawa et al. 2017[16]. Typically, females did not display attack bites[4] unless they were single-housed following Oliveira et al. 2021 (Supplementary Fig. 1, Supplementary video). Therefore, we scored all the non-aggressive behaviors mentioned above for males. However, aggressive/dominant behavior consisted of the percentage of time spent on threatening, pushing/shoving, mounting, chasing, and, eventually, biting the intruder. As virgin female mice do not easily display aggression, we used the social isolation and aggression training *(IST)* protocol described by Oliveira et al. 2021 to enhance the natural levels of aggression displayed by females. Briefly, IST was followed as described elsewhere[51]. Female C57BL6/J mice (8–12 weeks) were single-housed for 3 weeks and after social isolation, confronted with a same-sex and juvenile intruder for 4 consecutive days in the FIT (Fig. 2A).

Maternal aggression was measured using the *maternal defense test* as described elsewhere[37]. In brief, females were paired with experienced males for 1 week, and vaginal plugs were assessed to confirm pregnancy. Maternal aggression was tested by introducing a virgin female into the lactating female's home cage between lactation day 4 (LD4) and LD7, for 10 min.

### Sexual behavior

Sexual behavior was measured as previously described[25,36]. Briefly, Sexual behavior was videotaped and scored live. Tests also took place in the early hours of the dark phase under dim red light conditions. Typically, for male sexual behavior, male mice were paired with a receptive (estrus) naturally cycling C57BL6/J mouse in their home cage and left free to interact for 10 or 30 minutes, depending on the test day (Please see Figs. 2 and 3). Latencies for the first mount and first intromission as well as frequencies of mounts and intromissions were scored. The female estrous cycle was monitored via vaginal smears and histology. Briefly, vaginal washouts were observed under the microscope and classified in one of the four phases of the rodent estrous cycle: proestrus (when mostly epithelial cells were found), estrus (when mostly cornified cells were found), diestrus, and metaestrus (when leucocyte

infiltrates were found). For a detailed explanation of each one of those phases, please see refs. 11,48,52,53. Additionally, receptivity was confirmed prior to the experiment by pairing the estrous females with experienced males.

Female sexual behavior was assessed via the lordosis quotient (%). For neuronal ablation, experimental females had their ovaries removed via ovariectomy (under isoflurane anesthesia) and were implanted with a Silastic capsule (5 mm-long; inner diameter: 1.57 mm; outer diameter: 2.41 mm) containing crystalline 17-β-estradiol (0.14 µg/µl, E8875, Sigma, diluted 1:1 with cholesterol) subcutaneously (s.c.), followed by an injection with 500 µg progesterone (5 µg/µl, Sigma, P0130) (s.c.) 3 h prior to the sexual behavior test. This protocol was chosen to enable optimal levels of sexual receptivity in order to evaluate neuronal ablation effects. For neuronal activity experiments, naturally cycling and receptive females were tested. Importantly during the sexual behavior test, females were paired with experienced males, and the amount of lordosis and rejections displayed in response to male mount attempts was annotated. The test was terminated after 10 mounts or 10 min whatever came first.

## Mate and social preference
Mate and social preferences were assessed using a 3-chamber apparatus[12,54–57]. Briefly, the test consisted of 3 phases (each of which lasted 10 min). Habituation consisting of free investigation of the 3-chamber apparatus. Whereas during social preference experimental animals had to choose between an object and an adult unknown same-sex-conspecific, preference was calculated as an index of time investigating the conspecific/(time spent sniffing the conspecific+object). Directly after the social preference test, animals were given the option to choose between a same- and opposite-sex conspecific (mate preference), and preference was calculated as an index of time spent investigating the opposite-sex target/(time spent sniffing the same+opposite sex). Preference was analyzed using a one-sample t-test, and animals were assumed to prefer once their index differed from chance levels (0.5). Mice that did not leave the center chamber or spent more than 5 minutes in the center chamber were excluded from the analysis. Using this criterion, 1 male and 5 female mice were excluded from those analyses.

## Experiment overview
Experiment A (Fig. 2) Neuronal activity upon social encounter. Mice of both sexes were confronted either with an object (control) or an opposite-sex-conspecific for 15 min (mating). One hour later, those animals were again exposed to an object (controls) or a same-sex-conspecific (fighting). Importantly, males were confronted with smaller adult male mice whereas females were confronted with juvenile females[16] for 10 min (Fig. 2A). Directly after the last interactions, animals were anesthetized with ketamine (80 mg/kg) and xylazine (1 mg/Kg) solution, transcardially perfused with 4% paraformaldehyde (PFA) and had their brains collected and processed for immunohistochemistry targeting neuronal activity markers with different expression/phosphorylation dynamics, e.g., c-FOS for mating and pERK for fighting. Of note, both markers are known to be triggered by social interactions[15,16,36,58,59] including sexual and aggressive behavior, similar approaches have been used by others[60,61]. Importantly, only males that mounted during sexual behaviour and attacked were included in the neuronal activity analysis. Regarding females, only females who showed lordosis and dominant behaviors (described in detail in section 2.2) were included in the analysis).

Lactating[37] and socially isolated and aggression-trained (IST) females[4,38,49,51] were used as high female aggression controls. For IST females, two cohorts were trained in this schedule; the first one was used to confirm aggression/dominant behavior levels and detailed behavioral analysis as well as activation of the oxytocin system (Supplementary Fig. 1), as oxytocin has been shown to promote aggression in females[37,38,49,50]. The second cohort of IST females was confronted with a juvenile and perfused directly after behavioral testing to verify pERK phosphorylation of VMHvl-nNOS neurons upon fighting (Fig. 2A).

As lactating females are known to exhibit high levels of aggression[4,37], they were confronted between LD 04-07 with an adult virgin female and perfused directly after the maternal defense test to check for pERK phosphorylation after fighting (Fig. 2A).

## Experiment B (Fig. 3) VMHvl-nNOS ablation in male mice.
Male nNOS::Cre mice underwent stereotaxic surgery and were injected into the VMHvl bilaterally with either AAV1-flex-taCasp3-TEVp (ablation, $6.4 \times 10^{13}$ genomic copies per ml) or AAV1-flex-EF1a-EGFP-WPRE (control, $7.7 \times 10^{13}$ genomic copies per ml) (Supplementary fig. 2) virus, both virus were manufacture at the GIGA-Viral Vectors plataform at the University of Liege. After 3 weeks, animals underwent behavioral testing to measure different aspects of social behavior. First, sexual behavior was assessed by exposing experimental males to receptive females (in their home cage) for 10 min on 3 consecutive days. Directly after mating animals were then confronted with a male in their home cage for 10 minutes, sexual behavior testing prior to aggression was used to induce male territoriality[33,50]. Additionally, 5 days after the last RI, males were tested in a 30 min mating test,. Finally, 4 days after sexual behavior testing, animals underwent the 3-chamber test to measure social and mate preference. After behavioral testing animals were transcardially perfused with 4% PFA, and brains were harvested and used to confirm neuronal ablation via immunohistochemistry (Fig. 3A).

## Experiment C (Fig. 4) VMHvl-nNOS ablation in female mice.
Female nNOS::Cre mice (8–12 weeks) underwent stereotaxic surgery and were injected into the VMHvl bilaterally with either AAV1-flex-taCasp3-TEVp (ablation) or AAV1-flex-EF1a-EGFP-WPRE (control) virus (Supplementary fig. 2). After 3 weeks, animals underwent behavioral testing to measure different aspects of social behavior. First, all-natural cycling (intact) female mice underwent the 3-chamber test to measure social and mate preference. After the 3-chamber test took place, subjects were split into two cohorts. Cohort A was ovariectomized and implanted with a Silastic capsule containing crystalline 17-β-estradiol (diluted 1:1 with cholesterol) subcutaneously, followed by treatment with 500 µg progesterone 3 h prior sexual behavior test. Females were tested 3 times with a 4-day interval in between sexual behavior sessions. Cohort B remained ovary intact and underwent 3 FITs in 3 consecutive days. Finally, after the behavior tests were completed, mice from both cohorts were transcardially perfused and brains were harvested and used to confirm neuronal ablation via immunohistochemistry (Fig. 4A).

## Experiment D rescue
nNOS::Cre mice underwent stereotaxic surgery and were injected into the VMHvl bilaterally with the AAV1-flex-taCasp3-TEVp virus. After 3 weeks, males underwent sexual and aggressive behavior testing whereas females were only tested for sexual behavior. To try to rescue these behavioral impairments we used a combination of NO donor (SNAP) and a guanylate cyclase activator (BAY) 2 h before the behavioral testing in a within-subjects design[22,25]. Therefore animals were tested for each behavior three times, once to establish baseline levels of behavior and counterbalance drug and vehicle groups twice with intraperitoneal (i.p.) injections.

## Stereotaxic surgery
Stereotaxic surgery was performed under semi-sterile conditions[25,35]. nNOS::cre mice were anesthetized with a mixture of isoflurane (3–4% for initial anesthesia followed by 1–2% for sustained anesthesia), injected i.p. with the analgesic Temgesic (0.05 mg/kg Buprenorphine), and mounted in the stereotaxic frame. Then, a small incision was made in the skin to expose the skull. Bregma was used as a landmark to find the VMHvl (AP: -1.3; ML: ±0.6; DV: +5.7). After drilling the skull surface animals were infused with 1 µl of virus per side (0.1 µl/min). Following injection, the syringe was kept in place for at least 10 min to allow viral diffusion. Thereafter, the incision was sutured, and mice were transferred to their homecage and watched until they were fully awake. The animal's health

status was monitored periodically for the next 2 days, and behavioral tests started at least 3 weeks after surgery.

## Pharmacology

Mice were injected intraperitoneally with a cocktail of S-NITROSO-N-ACETYL-DL-PENICIL LAMININE (SNAP, N3398, Sigma-Aldrich) and BAY 41-2272 (BAY, B8810, Sigma-Aldrich), in a dose of 8 mg/Kg and 10 mg/Kg, respectively 2 h prior behavioral testing[25].

## Immunohistochemistry

After perfusion with 4% PFA, brains were frozen in dry ice and cryo-cutted. Slices (40 μm) containing the target regions were collected in cryoprotectant solution and stored at −20 °C until the experiment took place. A series of 6–8 slices were used for immunostaining. For immunohistochemistry (neuronal ablation confirmation), slices were washed in 0.1 PBS, and incubated in a solution of 3% $H_2O_2$ and 10% methanol for 15 min at room temperature (RT). Next, slices were washed with PBS 0.1 M and blocked for 1 h in blocking solution (Nomal goat Serum 5% + PBS + 0.5% Triton X-100 - PBST). Directly after blocking, slices were incubated in primary antibody rabbit-anti-nNOS antibody (1:1000, Invitrogen, #SAB4502010) at 4 °C overnight. Next, brain sections were washed in PBS 0.1 M, and incubated with the secondary antibody Bio-tin-SP AffiniPure-Goat-Anti-Rabbit (1:250; #111-065-003, JacksonImmunoResearch). After incubation, slices were washed in PBS, incubated for 1 h in the avidin-biotin (ABC kit, PK-4000, Vector Labs), washed in PBS, and incubated for 5 min in a gray substrate for peroxidase solution (SK-4700, Vector labs). Finally, sections were rinsed in 0.1 PBS and mounted on adhesive microscope slides (Superfrost Plus, Thermo Fisher Scientific), left to dry overnight, dehydrated in xylene, and mounted in Eukit (03989, Sigma, Aldrich).

For immunofluorescence (neuronal activity and ERα-nNOS colocalization), brain sec-tions were washed in 0.1 PBS, incubated in a solution of 3% $H_2O_2$ and 10% methanol at 15 min at room temperature (RT), then rinsed in Glycine buffer (0.1 M in PBS) for 20 min. Afterward, slices were washed with PBS and blocked for 1 h in a solution of normal goat (NGS 5% 0.1 PBS with 0.3% triton-x 100). Directly after blocking, slices were incubated in primary antibodies mouse-anti-oxytoci-neurophysinI (1:1000, p38, Harold Gainer), guinea-pig-anti-cfos (1:1000, 226–308, Synaptic Systems), rab-bit-anti-c-FOS (1:2000, ab190289, Abcam), (rabbit-anti-pERK anti-body (1:250, #4370, CellSignalling), goat-anti-nNOS (1:250, OSN00004G, Invitrogen), and rabbit-anti-ERα (1:1000, 06-935, Milipore) at 4 °C. Of note, the combination of anti-nNOS and anti-ERα or anti-OXT and anti-c-FOS (rabbit) was incubated overnight (Fig. 2) whereas anti-pERK and anti-c-fos were incubated together for 64 hours, thus anti-nNOS was incubated after the first round of secondary antibodies (targeting the primary antibodies for pERK and c-FOS) in an extra staining (Fig. 2). After primary incubation, slices were washed in PBS and incubated for 2 hours in RT with secondary antibodies nNOS-ERα (Alexa-fluor 546 goat-anti-rabbit and Alexa-fluor 488 donkey-anti-goat 1:250, Thermofisher) for pERK-c-Fos-nNOS (Alexa-fluor goat-anti-guinea-pig 488, goat-anti-rabbit 546 antibody and Alexa-fluor donkey-anti-goat 633, Thermofisher). After secondary antibody incubation, brain sections were rinsed in 0.1 PBS and mounted (Aqua-Poly mount medium, #18606-20) on adhesive microscope slides (Su-perfrost Plus, Thermo Fisher Scientific Inc, USA). Slides were kept in the dark at 4 °C until imaging.

Fluorescence slides were processed using an inverted confocal laser scanning micro-scope (Leica SP5, Leica Microsystems, GIGA-Cell imaging platform). Immunoperoxidase slides were imaged using an Epifluorescence Microscope (Echo Revolve, GIGA-Cell imaging Platform). Digital images were processed (Merging and Z-projections) using the Leica Application Suite X (Leica) and Fiji70. Cell counting was done by an experienced observer blind to the treatments (17).

Regarding data analysis neurons were counted in at least 4 sections per animal and averaged over animals. We analyzed sections from AP −1.4 until −1.8, importantly we collected 4 series of slices meaning that there was at least 160 μm between the first and fifth sections in the same series.

## Statistical analysis

Normality was tested using the Kolmogorov-Smirnov test. Once normality was found data was analyzed using a Student's t-test, Pearson's correlations, and Two-way ANOVAs. Once normality was not reached Mann-Whitney and Spearman correlations were performed. For detailed statistics please see Supplementary Tables 1–4.

## Statistics and reproducibility

Sample size was determined to be adequate based on the magnitude and consistency of measurable differences between groups and/ or conditions, using power analysis. Regarding replication, experiments A, B, and C used a single cohort. However, for the female aggression experiment and deletion of nNOS, multiple cohorts of mice were used. Importantly, neuronal as well as behavioral phenotypes regarding those experiments (Female aggression and nNOS deletion) have been replicated in different cohorts of animals. For detailed information please see the Material and Methods as well as Supplementary information.

## Reporting summary

Further information on research design is available in the Nature Portfolio Reporting Summary linked to this article.

## Data availability

The single data values used to build the figures can be found in an excel file (Supplement Data). Additional data, such as raw data, images, behavioral videos are available upon reasonable request made directly to the corresponding authors.

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

## Acknowledgements
We would like to thank Laura de Vries, Chloé Beaudou, and Lozen Thies for their help with the experiments and Dr. Harold Gainer for giving us the anti-oxytocin antibody. This work was supported by the Fonds de la Recherche Scientifique -FNRS Chargé de Recherche (1.B.308.23AGGRESSIONKiNG project to V.O., by the Fondation Léon Fredericq Bourse de fonctionnement Post-Doctorant (Neuromodulation of aggression by kisspeptin, neurokinin B, and GnRH to V.O., and the FNRS (PDR T.0015.20 to J.B.).

## Author contributions
V.O.: Designed, performed, analyzed experiments, and wrote the manuscript, as well as designed figures and tables. I.B.: Assisted with performing and analyzing behavioral experiments. J.B.: supervised the study and gave input to experimental design and to the manuscript. All authors contributed to the article and approved the submitted version.

## Funding

## Competing interests
J.B. is a Research Director of the FNRS, and she discloses a patent on J. Bakker and S. Ouerdi. "Agonists of human kisspeptin receptor for modulating sexual desire." WO2020/151830A1 filed -1 Unspecified -001 and issued 30 July 2020. All authors declare that the research was conducted in the absence of any commercial or financial relationships that could be construed as a potential conflict of interest. V.O. and I.B. declare no competing interests.
