## [Transparent Peer Review file · Communications Biology]

Ventromedial hypothalamus (VMHvl) nNOS neurons regulate social behaviors in a sex-specific manner

Corresponding Author: Dr Vinícius Oliveira

This manuscript has been previously submitted at another journal. This document only contains information relating to versions considered at Communications Biology.

Version 0:

Reviewer comments:

Reviewer #2

(Remarks to the Author)

The authors have sufficiently addressed the following reviewer comments listed below. However, in this reviewer's opinion, there are still some issues with the manuscript that have either been missed or not fully addressed since the original submission.

Reviewer 1 & 2 major concerns sufficiently addressed in resubmission:

- Reviewer 1 comment on possible inaccuracies between text and figure data on (nNOS-ERa)/ERa cell numbers. This reviewer does not find the numbers in Fig. 1 to contain discrepancies.
- Reviewer 1 comment about a lack of method details for immunohistochemistry experiments has been included in the supplemental methods section.
- Reviewer 1 comment on Figure 2, and the use of c-fos and pERK for labeling active neurons. This reviewer agrees with the authors response on the use of c-fos and pERK to label active neurons during different timeframes. This method is quite well established.
- Reviewer 1 comment asking for the rationale behind different behavioural designs between male and female mice. The authors have addressed this concern by adding the rationale to the text of the manuscript.
- Reviewer 2 comments 1 & 2 on the confusion of y-axis labels. The authors have addressed this point of confusion and now the figures and their measures are much clearer/easier to understand. However, the discrepancy in the data numbers under comment 2 does not seem to have been addressed (see below for more details).
- Reviewer 2 comment 3. The authors have addressed this and added a few sentences on ERa & nNOS in the introduction and discussion (although this may be better integrated into the text). Because the rationale for the experiment in Fig. 1 was not very clear in the text.

Major concerns remaining:

1. Supplementary Table 1 figure panels referenced are incorrect.

2. Section 2.2 needs to be rewritten. This section is very confusing for a multitude of reasons. (1) The title describes female specific aggression tests, but the text includes male aggression tests. (2) The way that it's written also makes it sound as though lactating females are different from those tested for maternal aggression (is this true?). (3) It doesn't include an explanation of IST but its procedures are later described in section 2.5. The distinction between these two sections (2.2 and 2.5) is unclear to me, and should be edited to be consistent, clear, and cohesive. (4) It's not clear what females were used in the Fig. 2A (left) experiments and what they were exposed to. Is the second paragraph of this section referring to the procedure of the Fig. 2A (left) females? (5) The content of this section jumps around and is not cohesive, making it difficult to understand.

3. Quantification of aggressive behaviours for the male and female data in Fig. 2 is not reported, even though it sounds like this data was collected in the methods. Perhaps none was collected for these animals, and if this is the case, this would be

an example of how confusing this reviewer found the methods text.

4. As Reviewer 1 previously suggested, Fig. 2 would benefit from either the graphs being edited to show total numbers of cells (rather than % of nNOS cells), or if the raw cell number data was included as a supplementary table (or figure). Relative measures such as % is ambiguous because differences can arise from changes in the numerator or denominator.

5. Figure 3 & 4 – there is a discrepancy in the numbers of animals that needs to be addressed. E.g., Fig 3. Some graphs have 5 control and 7 nNOS-del animals while others have 7 control and 9 nNOS-del. Supplementary table 2 & 3 numbers are also inaccurate because of this discrepancy. The authors need to explain why specific animals were excluded from their analyses. Same issues for Fig. 4. See 2 graphs for mate pref index and behavioural testing data vs. nNOS data. Authors should check all graphs and all data for accuracy, and all statistical tests for accuracy in the supplemental tables.

6. Figure 3 – discrepancy in mounts & intromissions data. I thank the authors for having clarified in the text a lot of the y-axes that were unclear in the original manuscript. However, the discrepancies in the data numbers still remain. E.g., Fig. 3E, I read mounts & intromissions as # of mounts + # of intromissions, but by my estimates, it is impossible to get some of these numbers for some of the data points presented in the mounts figure and/or intromissions figure. I considered maybe that mounts & intromissions could be interpreted as the intersection of mounts & intromissions, but this doesn't make sense either.

7. The rebuttal figure included for one of Reviewer 3 comments would be good to include in the manuscript (either in the main or supplemental figures).

Minor concerns

- Supplementary table 3 figure panels are out of order.
- Figure 4 legend, panel descriptions are out of order. A, B, D, C, E.
- Line 291: Wrong Figure #?
- Line 31-32: "expressed in a ratio of 1:1" does this mean ~100% of cells expressing PR also express ER? Ratio of 1:1 can mean different things (1:1 equal amounts of receptor in the same cell, or same number of cells express PR and the same number of different cells express ER). Perhaps the same clarification to line 455 would be helpful.

Reviewer #3

(Remarks to the Author)

I am satisfied with the responses to my queries. I will only suggest that the authors show the validation images provided in their rebuttal as part of the supplementary information

Version 1:

Reviewer comments:

Reviewer #2

(Remarks to the Author)

The authors have sufficiently addressed my previous concerns. The methods are now much clearer, the numbers of cell counts have been added to the supplementary tables, and errors corrected.

I ask that the authors please double check this last figure before publication:

Figure 3E - Is the scale on this figure correct? The numbers do not align with the tick marks on the graph.

Reviewer #3

(Remarks to the Author)

The authors have addressed my comments

Changes in the manuscript are highlighted in blue font

Reply: Reviewers' comments:

Reviewer #2 (Remarks to the Author):

The authors have sufficiently addressed the following reviewer comments listed below. However, in this reviewer's opinion, there are still some issues with the manuscript that have either been missed or not fully addressed since the original submission.

Reviewer 1 & 2 major concerns sufficiently addressed in resubmission:

- Reviewer 1 comment on possible inaccuracies between text and figure data on (nNOS-ERa)/ERa cell numbers. This reviewer does not find the numbers in Fig. 1 to contain discrepancies.
- Reviewer 1 comment about a lack of method details for immunohistochemistry experiments has been included in the supplemental methods section.
- Reviewer 1 comment on Figure 2, and the use of c-fos and pERK for labeling active neurons. This reviewer agrees with the authors response on the use of c-fos and pERK to label active neurons during different timeframes. This method is quite well established.
- Reviewer 1 comment asking for the rationale behind different behavioural designs between male and female mice. The authors have addressed this concern by adding the rationale to the text of the manuscript.
- Reviewer 2 comments 1 & 2 on the confusion of y-axis labels. The authors have addressed this point of confusion and now the figures and their measures are much clearer/easier to understand. However, the discrepancy in the data numbers under comment 2 does not seem to have been addressed (see below for more details).
- Reviewer 2 comment 3. The authors have addressed this and added a few sentences on ERa & nNOS in the introduction and discussion (although this may be better integrated into the text). Because the rationale for the experiment in Fig. 1 was not very clear in the text.

Reply: We thank the reviewer for recognizing our efforts to address the issues raised by reviewers 1 and 2. We also apologize that some information remains unclear.

Major concerns remaining:

1. Supplementary Table 1 figure panels referenced are incorrect.

Reply: We apologize for having overseen this mistake, which we have now fixed in the latest version of the Supplementary Information.

2. Section 2.2 needs to be rewritten. This section is very confusing for a multitude of reasons. (1) The title describes female specific aggression tests, but the text includes male aggression tests. (2) The way that it's written also makes it sound as though lactating females are different from those tested for maternal aggression (is this true?). (3) It doesn't include an explanation of IST but its procedures are later described in section 2.5. The distinction between these two sections (2.2 and 2.5) is unclear to me, and should be edited to be consistent, clear, and cohesive. (4) It's not clear what females were used in

the Fig. 2A (left) experiments and what they were exposed to. Is the second paragraph of this section referring to the procedure of the Fig. 2A (left) females? (5) The content of this section jumps around and is not cohesive, making it difficult to understand.

Reply: Our aim in section 2.2 was to describe how the behavioral tests to assess aggressive behavior were conducted generally, i.e., what parameters were scored, which type of intruders were used, and in what conditions animals were tested. Whereas in section 2.5 we wanted to give the authors an overview of how experimental procedures were conducted for a specific experiment such as in experiment 2 for neuronal activity analysis, going from behavioral testing to perfusion. However, we recognize that this might have caused confusion; therefore, we have decided to call section 2.2 only Aggressive behavior and describe each behavioral test in a specific paragraph. We hope those changes take away the reviewer's concerns.

3. Quantification of aggressive behaviours for the male and female data in Fig. 2 is not reported, even though it sounds like this data was collected in the methods. Perhaps none was collected for these animals, and if this is the case, this would be an example of how confusing this reviewer found the methods text.

Reply: Indeed, behavioral data were collected but not shown in the figures. We added this information to section 2A.

4. As Reviewer 1 previously suggested, Fig. 2 would benefit from either the graphs being edited to show total numbers of cells (rather than % of nNOS cells), or if the raw cell number data was included as a supplementary table (or figure). Relative measures such as % is ambiguous because differences can arise from changes in the numerator or denominator.

Reply: Based on the reviewer's concerns, we reanalyzed the data of Figure 2A, and we actually found out that males, but not females, who were exposed to behavior tended to exhibit a reduction in the number of nNOS neurons in the VMHvl, what was surprising. This information will now be included and discussed in the manuscript. As expected, differences regarding co-localization remained for female sexual behavior and male aggression when cell counts were used; those differences were absent for males exposed to sexual behavior. Anyway, these data fit our other findings, as male sexual behavior was not strongly disrupted by nNOS neuron ablation. Please see page 8 (lines 341-348) and page 11 (line 467).

5. Figure 3 & 4 – there is a discrepancy in the numbers of animals that needs to be addressed. E.g., Fig 3. Some graphs have 5 control and 7 nNOS-del animals while others have 7 control and 9 nNOS-del. Supplementary table 2 & 3 numbers are also inaccurate because of this discrepancy. The authors need to explain why specific animals were excluded from their analyses. Same issues for Fig. 4. See 2 graphs for mate pref index and behavioural testing data vs. nNOS data. Authors should check all graphs and all data for accuracy, and all statistical tests for accuracy in the supplemental tables.

Reply: We apologize for not specifying it better. Regarding Figure 3, we believe that the reviewer refers to the difference between the number of animals used for nNOS neuron cell counts and behavior. Indeed, not all animals used for behavior were used for histological confirmation; we have now made that clearer in the supplementary table 2.

The same applies to Figure 4; however, here we also split animals after mate and social preferences into two distinct cohorts, please see Figure 4A and Experiment C in the methods section. We have now made this info also clearer in supplementary table 3. Additionally, we have mentioned exclusion criteria for the mate and social preference tests in section 2.4. We also added the number of excluded animals (**lines 154-155**). We hope these changes take away the reviewer's concerns.

6. Figure 3 – discrepancy in mounts & intromissions data. I thank the authors for having clarified in the text a lot of the y-axes that were unclear in the original manuscript. However, the discrepancies in the data numbers still remain. E.g., Fig. 3E, I read mounts & intromissions as # of mounts + # of intromissions, but by my estimates, it is impossible to get some of these numbers for some of the data points presented in the mounts figure and/or intromissions figure. I considered maybe that mounts & intromissions could be interpreted as the intersection of mounts & intromissions, but this doesn't make sense either.

Reply: We appreciate that the reviewer acknowledges our efforts to respond to her/his comments. As we present individual plots for mounts and intromissions anyway, we decided to remove this specific plot pooling both behaviors from Figure 3E in order to avoid confusion. We hope that will answer the reviewer's questions.

7. The rebuttal figure included for one of Reviewer 3 comments would be good to include in the manuscript (either in the main or supplemental figures).

Reply: As suggested by the reviewer, the rebuttal figure is now included in the manuscript as supplementary figure 2.

Minor concerns

- Supplementary table 3 figure panels are out of order.

Reply: Indeed, panels do not follow figure structures, but we opted to keep this structure to keep similar animal numbers together. We hope that the reviewer respects our choice.

- Figure 4 legend, panel descriptions are out of order. A, B, D, C, E.

Reply: Indeed, we described the mate preference (panel C) again together with the sexual behavior from our point of view since these behaviors belong conceptually together. Therefore, we described them in the legend together. We hope that the reviewer can understand our point of view.

- Line 291: Wrong Figure #?

Reply: We apologize for overlooking this mistake and we corrected in the latest version of the manuscript.

- Line 31-32: "expressed in a ratio of 1:1" does this mean ~100% of cells expressing PR also express ER? Ratio of 1:1 can mean different things (1:1 equal amounts of receptor in the same cell, or same number of cells express PR and the same number of different cells express ER). Perhaps the same clarification to line 455 would be helpful.

Reply: We understand the reviewer's point, but that is literally how the authors who evaluated for colocalization expressed this ratio in their papers. Please see the papers of

the David Anderson and Dayu Lin labs. Anyway, we replaced 1:1 to nearly 100% colocalization and hope that this clarifies the reviewer's concerns, **please see line 24**.

Reviewer #3 (Remarks to the Author):

I am satisfied with the responses to my queries. I will only suggest that the authors show the validation images provided in their rebuttal as part of the supplementary information

Reply: We thank **Reviewer 3** for recognizing the potential of our manuscript and we would like to refer it to **comment number 7** of **Reviewer 2**.